

# Combining machine learning and SMILEs to classify, better understand, and project changes in ENSO events

Nicola Maher[1,2], Thibault P. Tabarin[3], and Sebastian Milinski[1,4,5]

[1]Max Planck Institute for Meteorology, Hamburg, Germany
[2]Cooperative Institute for Research in Environmental Sciences (CIRES) and Department of Atmospheric and Oceanic Sciences (ATOC), University of Colorado at Boulder, Boulder, CO 80309, USA
[3]Freelancer, Boulder, CO 80303, USA
[4]Climate and Global Dynamics Division, National Center for Atmospheric Research, Boulder, CO 80307, USA
[5]Cooperative Programs for the Advancement of Earth System Science, University Corporation for Atmospheric Research, Boulder, CO 80307, USA

**Correspondence:** Nicola Maher (nicola.maher@colorado.edu)

**Abstract.**

The El Niño Southern Oscillation (ENSO) occurs in three phases: neutral, warm (El Niño) and cool (La Niña). While classifying El Niño and La Niña is relatively straightforward, El Niño events can be broadly classified into two types: Central Pacific (CP) and Eastern Pacific (EP). Differentiating between CP and EP events is currently dependent on both the method and observational dataset used. In this study, we create a new classification scheme using supervised machine learning trained on 18 observational and reanalysis products. This builds on previous work by identifying classes of events using the temporal evolution of sea surface temperature in multiple regions across the tropical Pacific. By applying this new classifier to seven single model initial-condition large ensembles (SMILEs) we investigate both the internal variability and forced changes in each type of ENSO event, where events identified behave similar to those observed. It is currently debated whether the observed increase in the frequency of CP events after the late 1970s is due to climate change. We found it to be within the range of internal variability in the SMILEs. When considering future changes, we do not project a change in CP frequency or amplitude under a strong warming scenario (RCP8.5/SSP370) and we find model differences in EP El Niño and La Niña frequency and amplitude projections. Finally, we find that models show differences in projected precipitation and SST pattern changes for each event type that do not seem to be linked to the Pacific mean state SST change, although the SST and precipitation changes in individual SMILEs are linked. Our work demonstrates the value of combining machine learning with climate models, and highlights the need to use SMILEs when evaluating ENSO in climate models due to the large spread of results found within a single model due to internal variability alone.

## 1 Introduction

Understanding El Niño Southern Oscillation (ENSO) diversity is important due to the differing teleconnections and impacts of different types of events (Capotondi et al., 2020, and refs therein). ENSO occurs in three phases: neutral, La Niña, and El Niño. While El Niño events occur with a wide range of spatial structures (Giese and Ray, 2011) they can be broadly classified into





two types, which differ in evolution, strength and spatial structure (e.g. Capotondi et al., 2015, 2020). These are eastern Pacific (EP) El Niños, and central Pacific (CP) El Niños. EP El Niños have warm sea surface temperature (SST) anomalies located in the eastern equatorial pacific, typically attached to the coastline of South America, while for CP El Niños SST, wind and

subsurface anomalies are confined to the central Pacific (Kao and Yu, 2009). EP events tend to appear in the far east Pacific and move westward, with CP events generally appearing in the eastern subtropics and central Pacific (Kao and Yu, 2009; Yeh et al., 2014; Capotondi et al., 2015). EP events occur from weak to extremely strong in amplitude, while CP events tend to be on the weaker side (e.g. Capotondi et al., 2020).

Previous studies have classified El Niños into EP and CP events, but are limited by uncertainty in both the observed data and

the classification method. Pascolini-Campbell et al. (2015) summerize 9 classification schemes (Ashok et al., 2007; Hendon et al., 2009; Kao and Yu, 2009; Kim et al., 2009; Kug et al., 2009; Ren and Jin, 2011; Takahashi et al., 2011; Yeh et al., 2009; Yu and Kim, 2010) applied to 5 different SST products and show that event classification is dependent on both the index and dataset used. They identify events that appear with greatest convergence across indexes and datasets to provide the most robust classification of observations to date. Such classification using multiple SST products can now be automated using

machine learning, which has the additional advantage of using multiple parameters across many dimensions to identify events. This technique creates a more complete classifier than previous studies, where classification schemes have focused on single metrics with defined thresholds or comparing multiple schemes and products by hand (Pascolini-Campbell et al., 2015). Here, we use supervised machine learning to build a new ENSO classifier. We then apply this classifier to climate models to identify events that resemble those found in the real world. This is relevant as we wish to study events similar to those that we observe.

Machine learning techniques such as classification are becoming more commonly utilised in the geosciences. The key challenges in applying machine learning in this field come from the difficulty in ground truthing data, interpreting physical results, auto-correlation at both spatial and temporal scales, noisy data with gaps that is taken at multiple resolutions and scales, as well as data that is uncertain sparse and intermittent (Gil et al., 2018; Karpatne et al., 2019; Reichstein et al., 2019). Machine learning, however, also provides new opportunities in the geosciences particularly when using climate models where

large coherent datasets exist (Barnes et al., 2020, 2019; Toms et al., 2020).

ENSO research can be furthered by applying machine learning techniques. Two studies use clustering to identify different types of ENSO event without needing any prior information. Wang et al. (2019) find 4 types of ENSO event in observations, while Johnson (2013) identify 9 separate tropical Pacific patterns, including CP and EP El niños and a range of mixed event types. Machine learning has additionally been used to improve ENSO forecasts. Ham et al. (2019) use a convolutional neural

network to forecast ENSO up to 1.5 years in advance, improving on the prior forecasts that could not be made more than a year in advance. Their forecasts are shown to be better than the state of the art with additional analysis confirming that this technique performs well for physically reasonable reasons. Other recent studies have shown a combination of machine learning tools can be used to improve on ENSO predictions and create a even more reliable ENSO prediction than a neural network alone (Guo et al., 2020). However, no studies to date have used supervised learning combined with observations to investigate ENSO.

There is ongoing debate about the cause of an observed increase in the frequency of CP events after the late 1970s (An and Wang, 2000). This was initially attributed to increased greenhouse gas forcing (Yeh et al., 2009). Other studies then



suggested that multi-decadal modulation of CP frequency by internal variability can explain this increase (McPhaden et al., 2011; Newman et al., 2011; Yeh et al., 2011; Pascolini-Campbell et al., 2015). However, a recent study again suggested that this observed increase can be linked to greenhouse gas forcing (Liu et al., 2017). Additionally, paleoclimate records have shown

that the current ratio of CP to EP El Niños is unusual in a multi-century context (Freund et al., 2019). Freund et al. (2019) use corals from 27 seasonally resolved networks and find that compared to the last 4 centuries the recent 30-year period includes fewer, but more extreme EP El Niños and an unprecedented ratio of CP to EP El Niños compared to the rest of the record. It is currently unresolved why these past changes seen in the paleoclimate record occurred.

        Whether both the frequency and amplitude of EP and CP events will change in the future is also strongly debated. Multiple

studies agree that projections are inconsistent across CMIP5 models (Ham et al., 2015; Chen et al., 2017; Xu et al., 2017; Lemmon and Karnauskas, 2019). These differences are suggested to be related to the central pacific zonal SST gradient (Wang et al., 2019). When subsetting for models that represent ENSO well, studies find an increase in EP variability (Cai et al., 2018, 2021). However, when using single model initial-condition large ensembles (SMILEs), models are again found to differ in their projections (Ng et al., 2021). ENSO precipitation projections are more robust across all models with multiple studies

projecting an increase in ENSO related precipitation variability under strong warming scenarios (Power et al., 2013; Cai et al., 2014; Chung et al., 2014; Watanabe et al., 2014; Huang and Xie, 2015; Yun et al., 2021).

        While there is a wealth of literature investigating ENSO, previous work has highlighted that due to its high variability longer equilibrated runs or SMILEs are needed to truly understand our observations of ENSO and to project future changes (Wittenberg, 2009; Maher et al., 2018; Milinski et al., 2020). Two separate studies show that two SMILEs cover the spread

of ENSO projections in CMIP5 models for both traditional ENSO indicies (Maher et al., 2018) and EP and CP events (Ng et al., 2021). This indicates that internal variability can explain a large fraction of the inter-model spread previously attributed to model differences. These results highlight the danger of using single runs of climate models to investigate projections of high variability phenomena such as ENSO, where differing results can be found between models due to undersampling internal variability. This further demonstrates the utility of SMILEs, which provide many realisations of the earth system and allow

scientists to investigate both the climate change signal as well as inherently complex and noisy systems such as ENSO (Maher et al., 2018).

        The aims of this paper are twofold. First, we create a new classifier using supervised machine learning combined with 18 observational and reanalysis products. This classifier has the advantage that it can learn both the spatial and temporal evolution of different events unlike previous studies that rely on pre-defined metrics and comparing multiple methods and products by

hand. Second, we apply the classifier to seven SMILEs to answer three questions. One, can SMILEs capture the observed CP and EP El Niños, and La Niñas? Two, can the observed increase in frequency of CP El Niños be explained by internal variability and what does this imply for future projections? Three, do we project forced changes in the amplitude, SST and precipitation patterns of each event type? By using this classifier combined with SMILEs we can now better understand observations and projections of ENSO in the light of internal variability.



## 2   Creating the classifier

The purpose of a classifier is to assign a class to an event. In this paper our intent is to label each year with one of four event types: central Pacific (CP) El Niño, eastern Pacific (EP) El Niño , La Niña (LN), and neutral (NE). We use supervised learning in this study. Supervised learning algorithms use a labelled dataset (e.g. observations) to create a classifier, which can then be applied to unlabelled data (e.g. climate model output).

There are 7 steps to creating a machine learning classifier:

1. **Data Collection**: the quality and quantity of the data used dictates how well the classifier performs

2. **Data Preparation**: choosing which features from the dataset will be given to the classifier, preprocessing the data, as well as well as splitting the data into training and evaluation sets, labelling the data

3. **Choosing a classifier**: different algorithms are suited to different purposes and data types

4. **Training the classifier**: using the training set to train the classification algorithm

5. **Evaluating the classifier**: using the evaluation set to assess how the classifier performs, one must define an appropriate scoring metric fit for purpose

6. **Hyper-parameter tuning**: tuning the classification algorithm parameters for performance

7. **Prediction**: using the classifier to make predictions from other datasets, at this point we have a tool that will allow us to classify and investigate ENSO events in climate model output

The following sections provide a description of the steps followed in this study. These steps are merged into sections as some steps are performed in unison and/or iterated through.

### 2.1   Data collection and preparation

Steps 1 & 2 data collection and data preparation are outlined in this section. In this study we use SST data due to its good temporal and spatial coverage over the tropical Pacific Ocean. We choose not to include other variables due to their shorter record lengths. Additionally by using SST alone we can independently assess the projected precipitation response in climate models during the prediction phase. We label each year of the observational dataset as EP, CP, LN or NE using results from previous studies (Table 1). Here, we use the most robust set of labels available for CP from Pascolini-Campbell et al. (2015).

Machine learning classifiers work best with large amounts of training and testing data. Unfortunately, although for an observed climate variable SST has a relatively long record, there are only 124 years of data, which include only 14 CP events and 20 EP events (Table 1). This is an inherent problem when using climate data that has been highlighted in much of the literature (e.g. Reichstein et al., 2019). As the data is time-dependent we cannot easily acquire more data. Due to the spatial correlations within the dataset typical machine learning augmentation methods, which would be used to artificially create additional data, are not appropriate. Instead in this study we use an unconventional augmentation method where we opt to use 18 observational





and reanalysis products (Table S1). Each product includes the same events with the same labels, however due to observa-
tional uncertainty, infilling methods, and the use of reanalysis models, the SST patterns are different (e.g. Deser et al., 2010;
Pascolini-Campbell et al., 2015; Huang et al., 2018). This unconventional data augmentation takes into account observational
uncertainty in addition to effectively augmenting the data we have to use in this study. Before training the classifier we create
anomalies by detrending the data (using a third order polynomial) and removing the seasonal cycle (mean of each month over
the entire time-series after detrending).

The next step in the data preparation process is to identify which features to use in the study. While La Niña, neutral and
El Niño events are fairly easy to define, separating El Niño events into CP and EP classes can be difficult. Previous studies
have identified that the Pacific SST in the predefined niño boxes and the evolution of SST over the event, typically from
October to March are the most important features for separating the two types of El Niño (e.g. Rasmusson and Carpenter,
1982). In this study we use the regions niño3E (5S-5N, 120-90W), niño3W (5S-5N, 120-150W), niño4E (5S-5N, 150-170W),
niño4W (5S-5N, 170-200W), and niño1.2 (0-10N, 80-90W) in each month from October to March as individual features to
train the classifier (i.e. 5 regions x 6 months = 30 features). We note that we tested multiple feature sets for performance (see
Supplementary Information 2: Supplementary Methods), but chose this set due to its high performance in the evaluation phase.

In machine learning, part of the dataset is used for training, while the rest of the data is kept aside for evaluation of the clas-
sifier performance. In the following section when evaluating how the classifier performs we keep the HadISST observational
dataset aside and use all other data in the training phase.

## 2.2 Choosing, training, evaluating, and tuning the classifier

Steps 3, 4, 5 & 6: choosing, training, evaluating and tuning the classifier are outlined in this section. We outline these steps
together as we iterate through the different steps to find the best performing classifier. There are a suite of algorithms typically
used in supervised learning classifiers. It is common practice to test all algorithms and see how they perform on a particular
dataset, given they all have different strengths and weaknesses. The 9 algorithms tested in this study are: (1) nearest neighbours,
(2) linear support vector machine, (3) radical basis function support vector machine, (4) decision tree, (5) neural network, (6)
adaboost, (7) naive bayesian, (8) quadratic discriminant analysis, and (9) random forest. We then use the python scikit-learn
package (Pedregosa et al., 2011) to train each of these algorithms on our SST dataset. To evaluate which algorithms perform
best we use three different scores.

The first is an accuracy score:

$$accuracy = \frac{TP + TN}{TP + TN + FP + FN} \tag{1}$$

where, $TP$ are true positives, $TN$ are true negatives, $FP$ are false positives and $FN$ are false negatives.

The second is a precision score for each of the event types ($i$) that we want to classify:

$$\text{precision}_i(P - i) = \frac{TP_i}{TP_i + FP_i} \tag{2}$$

In both cases a score of 1 is best, while a score of 0 is worst. Both accuracy and precision scores are calculated on the evaluation
dataset.



The third score is an accuracy cross validation score (CVS), which we use to test how robust our estimate is. Here, to test the classifier, the algorithm breaks the training data into $n$ smaller datasets (in this study we use $n = 5$). For $n = 5$ the algorithm retrains the classifier using four of the five smaller datasets and tests on the fifth. We then obtain 5 accuracy scores which are averaged to give an estimate of the models performance. The cross validation score function in scikit-learn uses a KFold Stratification to split the data into these smaller datasets (Pedregosa et al., 2011). This is designed to keep the distribution of classes similar in each set and keep sets the same size.

In this study we are particularly interested in correctly identifying CP and EP events. This means that our algorithm must have high precision scores for these two events. Based on the scores outlined above we evaluate all nine algorithms as well as ensemble classifiers that use multiple algorithms (see Supplementary Information 2: Supplementary Methods for details of all algorithms tested & Table 2). The best performing algorithm is an ensemble voting classifier, which utilises the strengths of three algorithms. These three algorithms are: a neural network, a random forest and a nearest neighbour. All three algorithms hyper-parameters are tuned for optimal performance (step 6). The three algorithms chosen, the hyper-parameters used after tuning and the scores after tuning, are shown in Table 2. The ensemble voting classifier uses the wisdom of the crowd, where all three algorithms vote to give the final outcome. We choose to use soft voting as it performs better in the evaluation stage (see Supplementary Information 2: Supplementary Methods). Soft voting predicts the class label based on the maximum of the sums of the predicted probabilities. Evaluation scores for the final classifier are found in Table 2. The higher scores of this final classifier compared to each individual algorithm demonstrates the utility of combining three algorithms into a ensemble voting classifier. This final classifier correctly identifies 12/13 CP, 20/20 EP, 19/26 LN and 63/64 NE events. Those not classified correctly are identified as NE, except for the incorrectly classified NE event, which is classified as CP.

Before using this classifier in step 7 to classify climate model output we perform one more set of tests based on the following limitation. A limitation of the original evaluation is the choice of training and evaluation sets. In our original choice, where we reserve HadISST for evaluation the same ENSO events are effectively included in both the training and evaluation sets. While this is the typical way to split the data when using data augmentation techniques, we additionally test the sensitivity to this choice. To do this we use the longer datasets (ERSSTv3b, ERSSTv4, ERSSTv5, HadISST, kaplan and COBE), which all cover the years 1896-2016 and separate the data so that each event is only included in the training or evaluation set.

To decide on how to separate the data into training and evaluation sets we use the python function train test split (Pedregosa et al., 2011) which shuffles the data then splits it into training and evaluation sets while preserving the percentages of events in both the training and testing data. We do this 10 separate times to get 10 sets of training and evaluation data and then compute the scores shown in Table 3. We additionally complete this split 100 times and manually choose 10 data splits that take CP and EP (the classes with the lowest numbers of events) from across the time-series, ensuring that not all events in the split come from the same part of the observational record.

We find that for all three classifiers, there is a range of accuracy that depends on how the training and evaluation sets are constructed. We find that depending on this construction all classifiers have the most trouble identifying CP events, as shown by the lowest precision scores. The difficulty in identifying CP events is due to the relatively low number of events in this category. While the precision scores can be quite low for some training/evaluation sets, other sets show high precision, suggesting that





the algorithm can perform reasonably well. This suggests that if the training and evaluation sets are constructed in an ineffective way and the training set does not include specific events vital for classification, then the testing result is poor. This is a problem that occurs due to our small data size. Based on these results when training the final classifier we choose to use all available data (including HadISST) so as not to lose information by excluding events. This final classifier is used in step 7 prediction to identify each type of ENSO event in climate model output.

### 2.3 Prediction

In machine learning, step 7: prediction refers to applying the trained classifier to other datasets. We apply the classifier trained on observed SST data to climate model output from 7 SMILEs. Besides having a large amount of events to classify, the advantage of using SMILEs is that we can assess how internal climate variability can affect what we observe in our single realisation of the world. In this study we use 5 SMILEs with CMIP5 forcing (historical and RCP8.5) and two SMILEs with CMIP6 forcing (historical and SSP370), all of which have more than 20 members (Table S2). We classify using the same features as in our training and evaluation data. We discuss the SMILE classification results in the following sections.

## 3 Application to SMILEs

### 3.1 Can SMILEs capture the observed CP, EP Niños and La Niñas?

We find CP and EP El Niños as well as La Niñas in all models, and identify known biases in their spatial patterns (Figure 1). The composite spatial patterns of SST in the peak ENSO season (November, December, January) for each different ENSO class in comparison to the HadISST composites are shown in Figure 1. CP events are visibly weaker than EP events in both the SMILEs and observations. The known bias of ENSO where SST anomalies extend to far to the west is also clearly visible in all SMILEs, although it is more prominent in some models than others. Both the CSIRO and IPSL-CM6A models show particularly strong SST biases in the west, although their presentation is spatially different. The CSIRO model has a peak SST anomaly too far into the western Pacific, while IPSL-CM6A peaks in the central Pacific.

The time-evolution of the SST anomalies is found to be different for each event type and appears to be important for classification (Figure 2). We find that the machine learning classifier does not just learn the SST pattern shown in Figure 1 and that CP events are weaker than EP events. It also correctly identifies the general pattern of how the SST anomalies evolve over the event, with CP El Niños initiated in the central Pacific, while EP El Niños begin in the eastern Pacific off the South American coastline and evolve into the central Pacific over the course of the event. This is supported by the fact that the classifier performs better when information about the time-evolution is included in the feature choice (see Supplementary Information 2.1: Choice of Features). The two CMIP6 models have smaller relative SST anomalies than observations, with the relative SST anomalies in the CMIP5 models more realistic in magnitude (assuming that the observed record is long enough to sample these events). Overall, all models bar CSIRO show both a realistic evolution of both EP and CP events as well as the differences between the two.



Due to the relatively short observational record, which has been shown to be insufficient to capture multi-decadal ENSO
variability (Wittenberg, 2009; Maher et al., 2018; Milinski et al., 2020) we compare the range of frequencies found across each
ensemble with observations (Table 4). We find that the observed frequencies of EP and CP El Niños and La Niña events are
within the SMILE spread for all models except CSIRO, which does not realistically capture EP El Niño or La Niña frequency.
Additionally, all GFDL-ESM2M ensemble members have a too high frequency of CP events and CanESM5 has a too low
frequency of La Niña events. We also consider the pattern correlation between EP and CP events as compared to the observed
pattern correlation to see how similar EP and CP Niños can look due to internal variability (Table 4). We find that the observed
pattern correlation is always captured within the ensemble range of all SMILEs. However, individual ensemble members can
have a large range of pattern correlations. This demonstrates that when using single realisations, a model may appear to not
well represent the distinct EP and CP patterns by chance rather than due to model deficiencies.

Given climate models have known ENSO biases, particularly in the location of SST anomalies along the equator, we ad-
ditionally classify by shifting the longitudes of the niño regions. This shift is defined as the difference in location between
the maximum variability between 5N and 5S in the Pacific Ocean in the observations and the maximum variability in each
individual SMILE (Table S5). We find that this does not significantly change the results in the previous paragraphs except for
CSIRO frequency where EP Niños and La Niñas are now more realistically represented. This method additionally does not
change the results for climate projections presented in the following sections (see Supplementary Information 3: Shifted niño
regions).

When comparing these results to previous work, different studies identify different models as more realistic or able to
represent different ENSO event types (e.g Xu et al., 2017; Capotondi et al., 2020; Feng et al., 2020; Dieppois et al., 2021).
From our comparison with observations we have assessed that most models capture the observed frequency of all event types
and the correlation between the EP and CP Niño patterns, and demonstrate that previous work could find a variety of results
for the same models due to the use of single realisations. We show that all models exhibit some SST bias, but that EP and
CP events are differentiated in the models for physically interpretable reasons. In this case the classifier identifies the spatial
pattern, evolution, and amplitude of the different event types. However, just because some observed quantities fall within the
SMILE range, does not mean each model does not have individual and differing biases in other quantities as presented by
Bellenger et al. (2014); Karamperidou et al. (2017); Kohyama et al. (2017); Cai et al. (2018, 2021); Planton et al. (2021).
Given the validation performed in this section we choose to include all models except CSIRO in the following analysis of
projections. This is because CSIRO presents consistent strong biases in the pattern, evolution and frequency of events.

### 3.2   Can the observed increase in frequency of CP Niños be explained by internal variability and what does this imply for future projections?

The frequency of CP events was observed to increase after the 1970s (An and Wang, 2000) leading to the question of whether
this was a forced change due to increasing greenhouse gas emissions (Yeh et al., 2009). In Figure 3 we show that the range
of frequencies in CP (and EP and La Niña) events across a individual SMILE for a 30-year period is large. This implies that
the observed increase in CP events could be due to internal variability alone, similar to Pascolini-Campbell et al. (2015) and





Dieppois et al. (2021), who pointed out that CP frequency varies on multi-decadal timescales. This is the case in the SMILEs, when we consider a single ensemble member (Figure 3; green lines) there are periods where it sits higher and lower in the
ensemble spread, demonstrating this multi-decadal variability.

When assessing projected changes in frequency, we use a signal to noise ratio to identify changes. When the signal at any time point (ensemble mean at the time point minus the ensemble mean at the beginning of the time-series) is greater than the noise (standard deviation taken across the ensemble) we identify a *likely* projected change. When the signal is 1.645 or 2 times the noise these thresholds correspond to the *very likely* and *extremely likely*. CESM-LE (and to some extent CanESM5 and IPSL-
CM6A) show a likely increase in EP El Niño frequency, with CESM-LE and CanESM5 and a likely to very likely increase in La Niña event frequency, with CanESM2 and GFDL-ESM2M showing likely decreases in La Niña event frequency respectively (Figure 3). No model projects a significant change in CP frequency. We note that large changes could be observed due to the considerable internal variability of these frequencies alone as demonstrated by the maximum and minimum frequencies and the multi-decadal changes seen in the individual ensemble members. These results are interesting in the context of a paleoclimate
study by Freund et al. (2019) who show that the current relative frequency of EP/CP events is unprecedented in the paleoclimate record. There are two possible reasons for the differences between this study and our results. First, model biases may mean that the models cannot capture this shift correctly. Second, a shift in frequencies may have already occurred, after which we do not project further future shifts. This could be investigated further in future studies by applying this classification method to the last millennium long paleoclimate model runs.

**3.3 Do we project changes in the amplitude, SST and precipitation patterns of each event type?**

When considering ENSO amplitude (November, December, January mean for the region 160E to 80W between 5N and 5S after the ensemble mean has been removed for each event), similar to frequency we find that the range of amplitudes across each SMILE at any given time is quite large ($0.5 - 1.5^oC$) for each event type (Figure 4). This agrees well with previous work that shows that ENSO amplitude is very variable in single realisations of climate models (e.g. Wittenberg, 2009; Maher et al., 2018;
Ng et al., 2021; Dieppois et al., 2021). When projecting future changes, we find model disagreement on the forced change in EP El Niño and La Niña amplitude. We find model agreement of no change in CP El Niño amplitude until right at the end of the time-series where CanESM2 and CanESM5 show significant decreases and increases in amplitude respectively (Figure 4). We find a likely increase in the amplitude of EP events in CESM-LE and a likely to very likely increase in CanESM5, however there is an extremely likely decrease in CanESM2 and GFDL-ESM2M. For La Niña events there is a likely decrease
in CESM-LE, and very likely increases in CanESM2 and GFDL-ESM2M.

These results compare well to previous work, where El Niño amplitude was found to increase in CESM-LE, not change in MPI-GE, and decrease in GFDL-ESM2M (Zheng et al., 2017; Haszpra et al., 2020; Maher et al., 2018, 2021; Ng et al., 2021). When partitioning amplitude changes into EP and CP events using different criteria for classification, Ng et al. (2021) found an increase in both EP and CP amplitude in CESM-LE and no change in MPI-GE. This agrees well with our results, although
we do not see the increase in CP amplitude in CESM-LE, possibly due to the different time periods used and the relatively small magnitude of the change. Recently Cai et al. (2018) and Cai et al. (2021) showed that for CMIP5 and CMIP6 models



that represent some ENSO properties well, EP amplitude increases in most models. However, only two of the models they identified as able to represent ENSO well are included in our study (CESM-LE and GFDL-ESM2M) and they differ in sign (increasing amplitude in CESM-LE; decreasing amplitude in GFDL-ESM2M). This is in agreement with Cai et al. (2018) who 290    also found decreasing amplitude in GFDL-ESM2M, which was an outlier in their study which used single ensemble members from CMIP5.

These model differences have been suggested to be related to changes in the zonal gradient of mean SST. Wang et al. (2019) suggest that an increase in the SST gradient results in an increase in the amplitude of strong basin wide El Niños, with a decrease leading to the a decrease in amplitude. Kohyama et al. (2017), however, suggest that a La Niña like warming pattern 295    (i.e. the western Pacific warms more than the eastern Pacific, resulting in an increased zonal SST gradient), should result in a decrease in ENSO amplitude as does Fredriksen et al. (2020). This result is in agreement with an observed decrease in tropical Pacific variability coupled with an increase in the trade winds and increase in thermocline tilt from 2000-2011 as compared to 1979-1999 (Hu et al., 2013). Beobide-Arsuaga et al. (2021) also find a weak negative correlation between the two quantities in CMIP5 and CMIP6 models. In our study three of the five SMILEs additionally show a weak negative relationship between 300    these quantities, however two models (CanESM2 and CanESM5) show no relationship (Figure S2a). We find no relationship between CP amplitude and the projected change in the zonal mean SST gradient (Figure S2b) consistent with Fredriksen et al. (2020).

We next plot the change in SST pattern for each event type for the period 2050-2099 as compared to 1950-1999 (Figure 5). We use detrended data to look at the changes in ENSO itself, outside of mean-state changes. We find similar pattern changes 305    for EP and CP events, with the opposite change for LN events. The SST pattern change lacks agreement across the models. Potentially this is related to the mean state changes (Knutson et al., 1997; McPhaden et al., 2011; Hu et al., 2013; Kim et al., 2014; Kohyama et al., 2017; Wang et al., 2019; Beobide-Arsuaga et al., 2021), however the SST pattern change is different for CESM-LE and CanESM2, which have similar mean state changes suggesting that the relationship is more complicated than one simple metric.

310    Precipitation projections for each event type also show large differences between SMILEs (Figure 6). These projected pattern differences are, however, found to be linked to the projected SST patterns, with contours of increasing SST found to closely correspond to wetting, and decreasing SST to drying. This clarifies that ENSO changes in SST and precipitation are linked and that it is not possible to truly understand one without the other.

We compare the multi-ensemble mean patterns (Figure 6) to previous work using CMIP models. Xu et al. (2017) find 315    similar strong cooling in RCP8.5 in the eastern Pacific for EP events, and some similarities of cooling in the eastern Pacific for CP events as well. For precipitation projections, EP events look similar to Xu et al. (2017) and show a general increase in precipitation, particularly in the central Pacific, however CP events look quite different. This is likely due to the definitions used by Xu et al. (2017) where they define EP events as the first empirical orthogonal function (EOF1) in tropical Pacific and CP as second empirical orthogonal function (EOF2). Other studies have, however, suggested that it is the combination of EOF1 320    and EOF2 that should be used to identify EP and CP events (Takahashi et al., 2011). When comparing CMIP5 projections from Power et al. (2013), who also look at EOF1, some similarities are found for precipitation projections, but SST projections look





quite different. These results likely differ both due to different definitions of event types as well as the different set of models used and differences between single model realisations and SMILEs. We note that we also consider extreme El Niños, and discuss results for this event type in Supplementary Information 4: Extreme Niños.

## 4    Summary and Conclusions

In this study we use supervised machine learning combined with SST products to develop a new classifier for ENSO events, which classifies events into La Niña, neutral, Eastern Pacific (EP) El Niño, and Central Pacific (CP) El Niño. This method uses differences in the pattern, amplitude, and evolution of events to make the classification. Using supervised machine learning has the advantage that it includes spatial and temporal information from 18 SST products to classify events without relying on pre-defined metrics, individual parameters, or manual identification. We then apply this classifier to seven SMILEs to identify ENSO events similar to those observed. By using SMILEs we examine forced changes in ENSO compared to the magnitude of internal variability. We find that:

1. All SMILEs bar CSIRO capture the observed pattern, evolution, frequency, and pattern correlation of EP and CP events, although known biases in the spatial pattern (e.g. SST anomalies too located too far west) are found.

2. The observed increase in the frequency of CP events is within the range of the SMILEs internal variability.

3. EP and CP El Niño frequency and CP amplitude are not projected to change in the future.

4. The SMILEs do not agree on projections of EP El Niño amplitude and La Niña frequency and amplitude. EP event amplitude projections are found to be weakly linked to changes in the zonal-mean gradient across the Pacific.

5. Models show differences in projected patterns of ENSO SST and precipitation that do not seem to be simply linked to the tropical Pacific mean state changes. However, the precipitation and SST changes in individual models are linked.

In conclusion, supervised machine learning has be used to build a new ENSO classifier for climate models that takes into account SST evolution along the tropical Pacific, and can be used to identify events that behave similarly to those observed. The models do not project changes in CP El Niño frequency or amplitude and demonstrate disparity in future changes in other event types, and in the projected spatial patterns of SST and precipitation. The large ensemble spread for frequency and amplitude highlights, similar to previous work, that SMILEs are are needed to evaluate ENSO and make projections due to the large variability of ENSO characteristics on decadal and longer timescales.

*Code and data availability.* The data that support the findings of this study are openly available at the following locations: MPI-GE, https://esgf-data.dkrz.de/projects/mpi-ge/, all other CMIP5 large ensembles (CanESM2, CESM-LE, CSIRO-Mk3-6-0, GFDL-ESM2M ); http://www. cesm.ucar.edu/projects/community-projects/MMLEA/ and CMIP6, https://esgf-data.dkrz.de/projects/cmip6-dkrz/. Source code for the machine learning classifier and the observed data which it is trained on can be found on github at: https://github.com/nicolamaher/ classification. Derived data supporting the findings of this study will be availalbe on final publication from https://pure.mpg.de/



*Author contributions.* NM and TT jointly devised the study. TT provided the machine learning expertise and NM provided the ENSO expertise. SM had the idea to use multiple SST products and reanalysis for data Augmentation. NM completed the analysis and wrote the manuscript with critical input from TT and SM.

355 *Competing interests.* The authors declare no competing interests.

*Acknowledgements.* We thank Jeff Exbrayat for his discussion on machine learning, Elizabeth Barnes for her comments on the project, including testing/training split, John O'Brien for his ideas on extreme ENSO section and Malte Stuecker for his comments on the 2019/2020 state of the tropical Pacific. We additionally thank Goratz Beobide Arsuaga for his internal review of this manuscript and Antonietta Capotoni for her comments on the manuscript. NM was supported by the Max Planck Society for the Advancement of Science and the CIRES Visiting
360 Postdoctoral Fellowship. SM was also supported by the Max Planck Society for the Advancement of Science. Finally, we thank Jochem Marotzke for his mentorship and support of this project. COBE, GODAS, KAPLAN and OISST data provided by the NOAA/OAR/ESRL PSL, Boulder, Colorado, USA, from their Web site at https://www.psl.noaa.gov/data/gridded/.





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



**Figure 1.** SST pattern for composites of EP and CP El Niños, and La Niña events (left, middle and right columns respectively). Shown for HadISST observations (top row) and each individual SMILE (in order of appearance; MPI-GE, CESM-LE, CanESM2, GFDL-ESM2M, CSIRO, CanESM5 and IPSL-CM6A). The SST pattern is shown for the November, December, January average. SMILE data has the forced response (ensemble mean) removed prior to calculation, HadISST is detrended using a second order polynomial then each months average is removed. The time period used is all of the historical, which is shown for the observations in Table S1 and SMILEs in Table S2.





**Table 1.** Years which are defined as Central Pacific (CP) and Eastern Pacific (EP) El Niños, La Niñas (LE), and Neutral (NE) events in the observational data. These years are found using Pascolini-Campbell et al. (2015) for CP years, https://psl.noaa.gov/enso/past-events.hptp for events 1896-2014 and https://origin.cpc.ncep.noaa.gov/products/analysis_monitoring/ensostuff/ONI_v5.php for events 2014-2019, and https://www.pmel.noaa.gov/tao/drupal/disdel/ to determine whether the 2019 El Niño was a CP event

| event type (total no. events) | years (starting year ) |
|---|---|
| CP (14) | 1914, 1940, 1958, 1963, 1968, 1977, 1986, 1990, 1991, 1994, 2002, 2003, 2004, 2019 |
| EP (20) | 1896, 1899, 1902, 1905, 1918, 1925, 1930, 1941, 1957, 1965, 1972, 1979, 1982, 1987, 1997, 2006, 2009, 2014, 2015, 2018 |
| LN (26) | 1903, 1908, 1909, 1910, 1916, 1917, 1924, 1933, 1938, 1942, 1949, 1950, 1954, 1955, 1961, 1970, 1973, 1975, 1988, 1998, 1999, 2007, 2010, 2011, 2016, 2017 |
| NE (64) | all others between 1896-2019 |

**Table 2.** Scores for different algorithms tested. Scores are defined in section 2.2.

| Algorithm | python name | tuned hyper-parameters | Accuracy | CVS | P-CP | P-EP | P-LN | P-NE |
|---|---|---|---|---|---|---|---|---|
| (1) NearestNeighbours | KNeighborsClassifier | n neighbors=1 | 0.93 | 0.89 | 0.92 | 1 | 1 | 0.90 |
| (5) NeuralNet | MLPClassifier | hidden layers=500 900 max iterations= 900 | 0.91 | 0.91 | 0.88 | 0.91 | 1 | 0.90 |
| (9) RandomForest | RandomForestClassifier | n estimators=500 max depth=500 | 0.88 | 0.89 | 0.79 | 01 | 1 | 0.84 |
| **FINAL: ensemble classifier** | VotingClassifier | Soft vote (1,5,9) | 0.93 | 0.92 | 0.92 | 1 | 1 | 0.9 |





**Table 3.** Minimum, mean and maximum scores for the final ensemble classifier. Test 1 uses all available data, with HadISST kept aside for testing. This test is run 10 times to find the minimum and maximum scores possible due to stochastic noise in the algorithms. Test 2 uses the longer datasets, ERSST, COBE, Kaplan and HadISST for training and testing. The data is split so that the augmented events must all occur in the same section of the data. The data split is randomly completed 10 times on alternative splits of the training and testing data. To complete this we use the python function *train test split*. Test2 w/check again uses the same function, however 10 data splits are manually chosen to ensure that they sample events from across the time-dimension and have a reasonable amount of each type of event.

| Test | Min/Max score | Accuracy | CVS | P-CP | P-EP | P-LN | P-NE |
|---|---|---|---|---|---|---|---|
| Test 1 | Min | 0.93 | 0.91 | 0.92 | 1 | 1 | 0.89 |
| | Mean | 0.93 | 0.92 | 0.92 | 1 | 1 | 0.9 |
| | Max | 0.93 | 0.92 | 0.92 | 1 | 1 | 0.9 |
| Test 2 random | Min | 0.65 | 0.93 | 0.08 | 0.23 | 0.70 | 0.61 |
| | Mean | 0.74 | 0.94 | 0.23 | 0.55 | 0.83 | 0.84 |
| | Max | 0.82 | 0.96 | 0.53 | 0.79 | 0.97 | 0.93 |
| Test 2 w/check | Min | 0.67 | 0.93 | 0.24 | 0.56 | 0.54 | 0.66 |
| | Mean | 0.73 | 0.94 | 0.50 | 0.67 | 0.82 | 0.77 |
| | Max | 0.81 | 0.96 | 0.77 | 0.97 | 1 | 0.89 |

**Table 4.** Frequency of events (as a percentage) in the historical period for observations (HadISST) and the SMILEs, as well as the correlation between EP and CP patterns. The mean frequency and correlation across each ensemble is shown with the minimum and maximum in brackets. The time period used is all of the historical, which is shown for the observations in Table S1 and SMILEs in Table S2.

| Model | EP no ev | CP no ev | LN no ev | EP/CP corr |
|---|---|---|---|---|
| HadISST | 16.1 | 11.2 | 21.0 | 0.85 |
| MPI-GE | 16.3 (9.7/22.6) | 10.9 (6.5/16.8) | 18.0 (8.4/24.5) | 0.82 (0.68/0.90) |
| CESM-LE | 22.1 (15.3/29.4) | 8.5 (3.5/15.3) | 25.0 (18.8/35.3) | 0.76 (0.56/0.88) |
| CanESM2 | 20.4 (14.3/27.1) | 12.2 (5.7/21.4) | 22.6 (12.9/31.4) | 0.84 (0.74/0.94) |
| GFDL-ESM2M | 16.2 (9.1/25.5) | 19.0 (12.7/25.5) | 27.0 (20/36.4) | 0.75 (0.64/0.88) |
| CSIRO | 9.4 (5.2/13.5) | 15.1 (7.1/21.3) | 13.7 (9.0/18.8) | 0.88 (0.80/0.95) |
| CanESM5 | 11.9 (7.3/17.7) | 7.1 (3.7/12.8) | 14.1 (9.8/18.3) | 0.83 (0.72/0.91) |
| IPSL-CM6A | 18.4 (12.8/22.6) | 9.8 (4.9/16.5) | 20.2 (15.9/23.8) | 0.85 (0.75/0.90) |

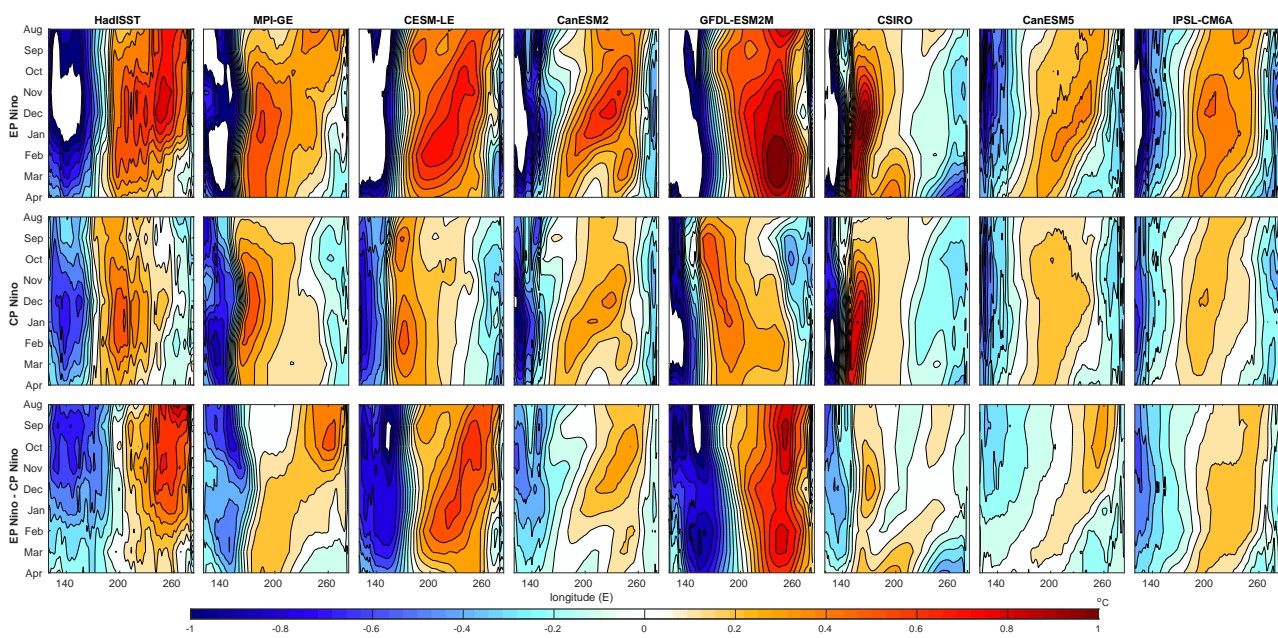

**Figure 2.** Hovmöller of relative SST along the equator in the Pacific Ocean for composites of EP and CP El Niños and EP-CP El Niños (top, middle and bottom row respectively). Shown for HadISST observations (left column) and each individual SMILE (in order of appearance; MPI-GE, CESM-LE, CanESM2, GFDL-ESM2M, CSIRO, CanESM5 and IPSL-CM6A). SST is averaged between 5N and 5S and shown for August to April. SMILE data has the forced response (ensemble mean) removed prior to calculation, HadISST is detrended using a second order polynomial then each months average is removed. The time period used is all of the historical, which is shown for the observations in Table S1 and SMILEs in Table S2. Relative SST is calculated by removing the average SST between 120E and 280E individually for each month.



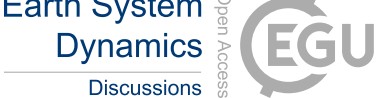

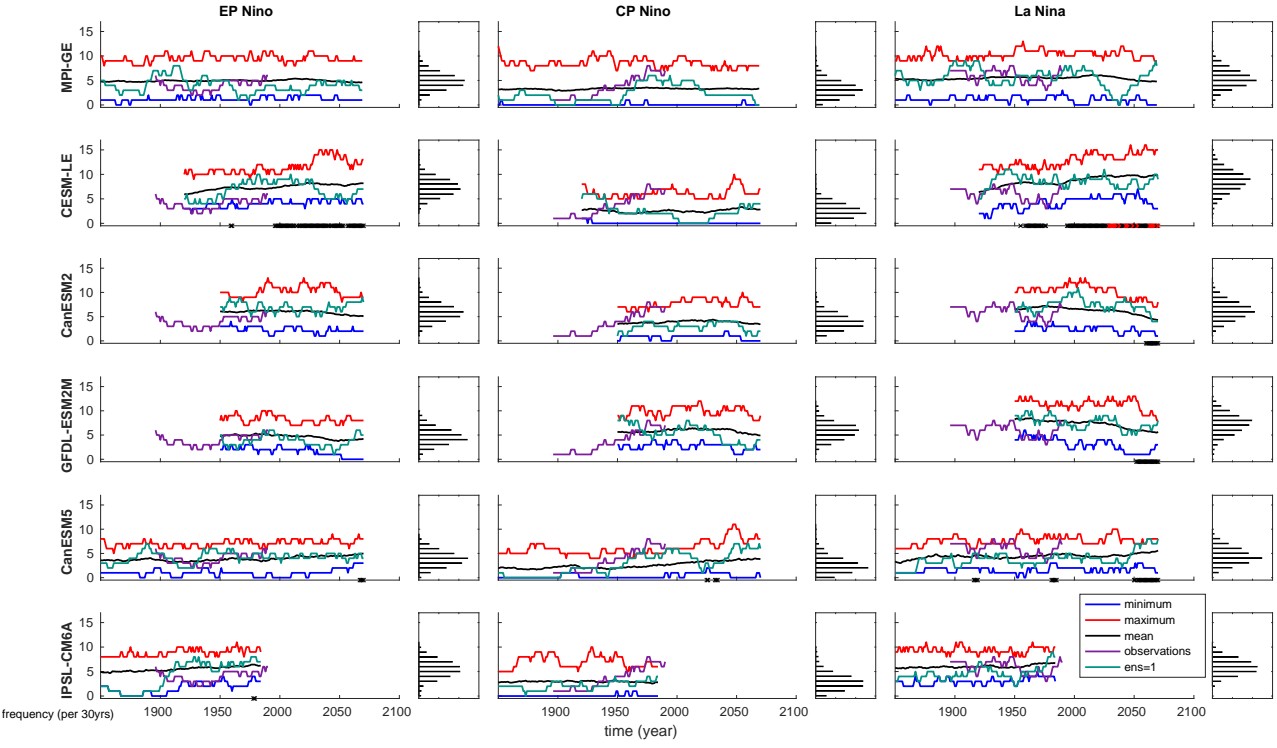

**Figure 3.** ENSO frequency in each SMILE for EP and CP El Niños, and La Niña events (left, middle and right columns respectively). Black line shows the ensemble mean for each year, red line shows the ensemble maximum, the blue line the ensemble minimum, the purple line is HadISST observations, and the green line is the first ensemble member. Frequency is calculated as the number of events in a single ensemble member per 30 years, taken as a running calculation along the time-series. PDFs show the distribution of ensemble members for the entire time-series. Black dots on the x-axis demonstrate when the signal (current ensemble mean minus the ensemble mean at the beginning of the time-series) is greater than the noise (standard deviation taken across the ensemble). Red dots show when the signal is 1.645 times the noise, while magenta dots show the same when the signal is greater than 2 times the noise. These thresholds correspond to the *likely*, *very likely* and *extremely likely* ranges.



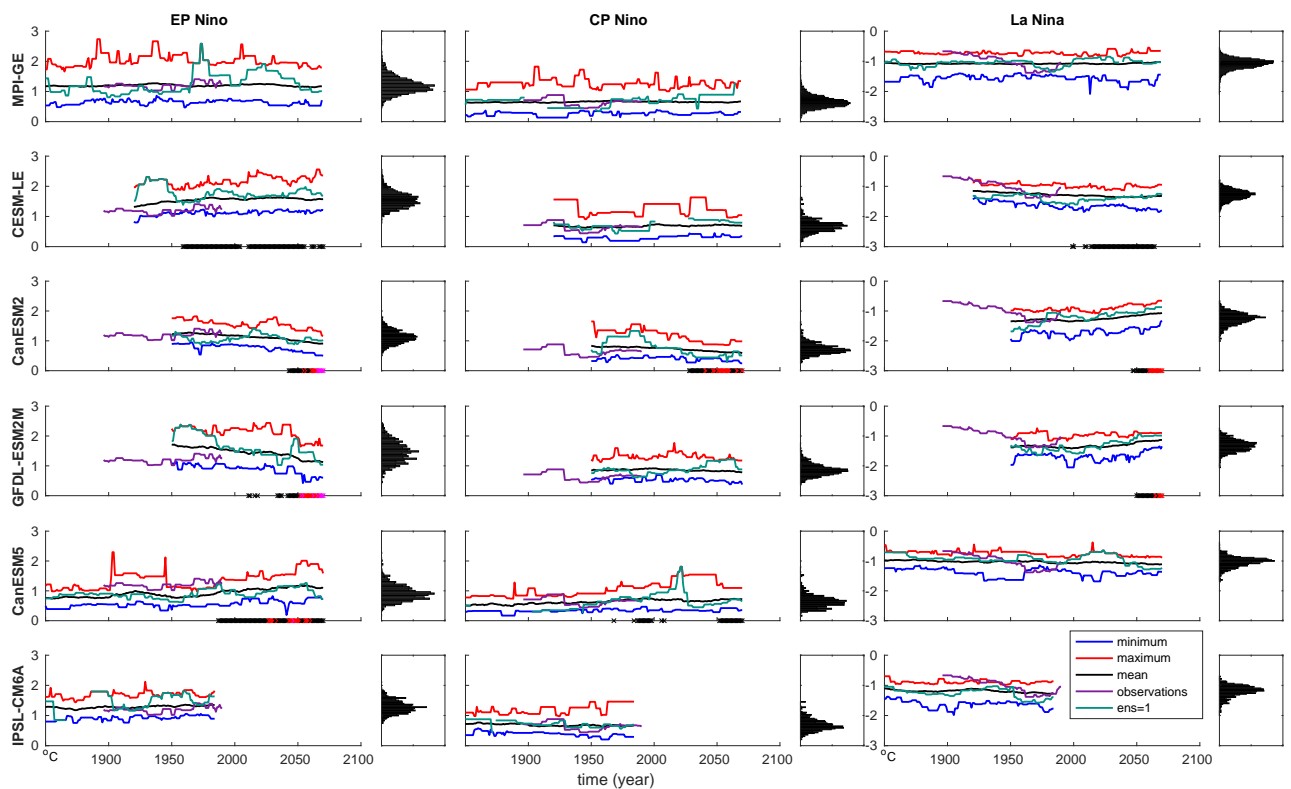

**Figure 4.** ENSO amplitude in each SMILE for EP and CP El Niños, and La Niña events (left, middle and right columns respectively). Black line shows the ensemble mean for each year, red line shows the ensemble maximum, the blue line the ensemble minimum, the purple line is HadISST observations, and the green line is the first ensemble member. Amplitude is calculated as the November, December, January mean for the region 160E to 80W between 5N and 5S after the ensemble mean has been removed for each event. PDFs show the distribution of ensemble members for the entire time-series. Black dots on the x-axis demonstrate when the signal (current ensemble mean minus the ensemble mean at the beginning of the time-series) is greater than the noise (standard deviation taken across the ensemble). Red dots show when the signal is 1.645 times the noise, while magenta dots show the same when the signal is greater than 2 times the noise. These thresholds correspond to the *likely*, *very likely* and *extremely likely* ranges.

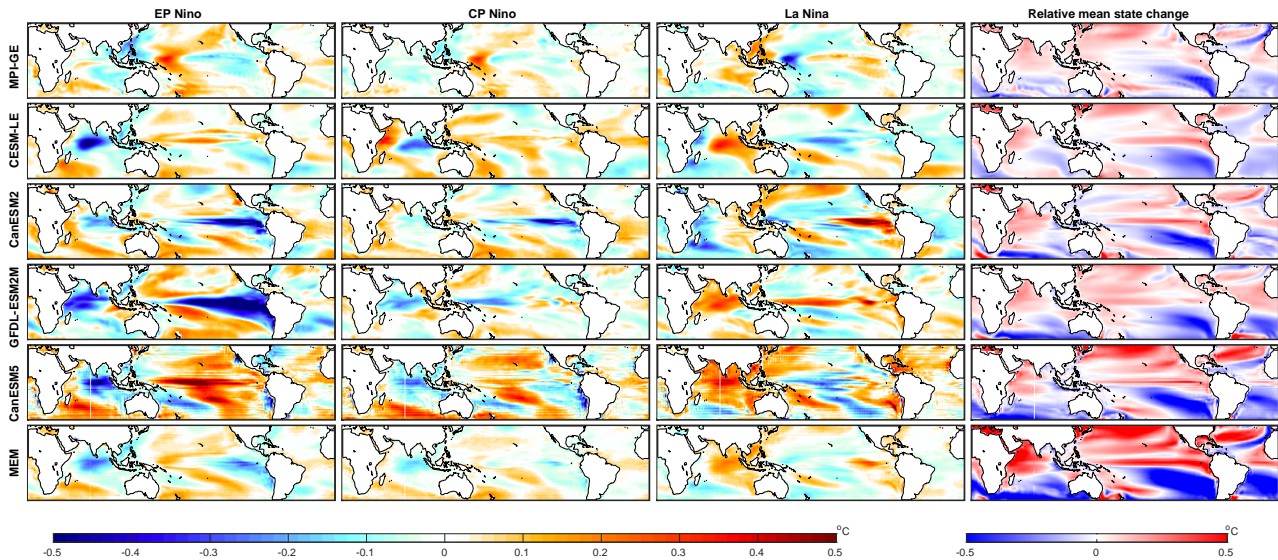

**Figure 5.** Change in SST pattern in each SMILE in the period 2050-2099 as compared to 1950-1999 for EP and CP El Niños, and La Niña events (left, centre left and centre right columns respectively. The mean state change in SST is shown in the right column. Shown for each individual SMILE (in order of appearance; MPI-GE, CESM-LE, CanESM2, GFDL-ESM2M, CanESM5) and the multi-ensemble mean (bottom row). The SST pattern is calculated as the November, December, January average and composited for each event type over each time-period. The relative mean state change is calculated as the ensemble mean SST over each time period, with the earlier period subtracted from the latter relative to the total change in the region shown (0-360E, 40S-40N). SMILE data has the forced response (ensemble mean) removed prior to calculation for the SST change, but not the mean state change.

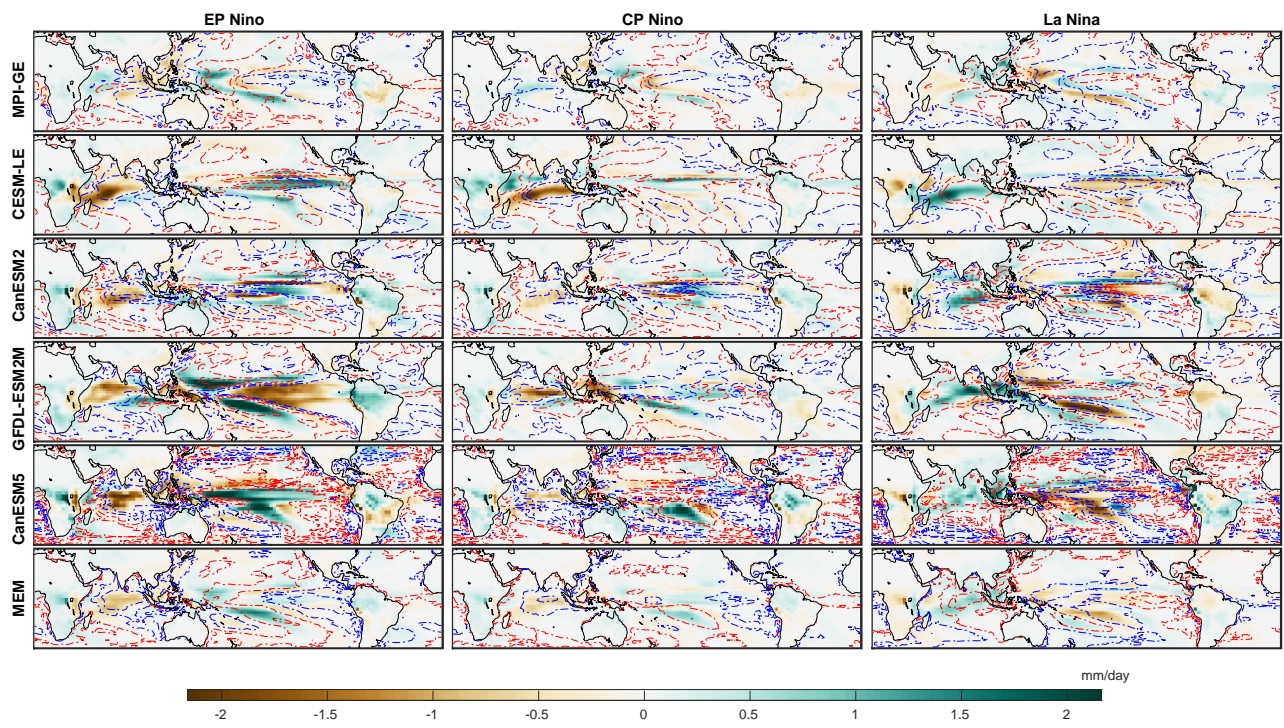

**Figure 6.** Change in precipitation pattern in each SMILE in the period 2050-2099 as compared to 1950-1999 for EP and CP El Niños, and La Niña events (left, middle and right columns respectively). The SST change from Figure 5 is shown as contours for reference (blue=cooling, red=warming). Shown for each individual SMILE (in order of appearance; MPI-GE, CESM-LE, CanESM2, GFDL-ESM2M, CanESM5) and the multi-ensemble mean (bottom row). The precipitation pattern is calculated as the November, December, January average and composited for each event type over each time-period. SMILE data has the forced response (ensemble mean) removed prior to calculation.