# Peer review of "Combining machine learning and SMILEs to classify, better understand, and project changes in ENSO events"

_Earth System Dynamics, 2021_

## Referee Comment (RC2)

This paper applied the integrated observational dataset to train the classification of the EP El Niño, CP type El Niño, and La Niña with supervised learning and to investigate the ENSO diversity/complexity changes in multi-model large ensembles. Specifically, they found the supervised machine learning can reasonably classify ENSO events/types and the observed increase of CP El Niño events is within the range of internal variability, so does the ENSO amplitude and frequency changes. The research topic is interesting and necessary; however, there are issues in the machine learning setup and the goal/finding is not unique for machine learning. Therefore, this paper should not be accepted in Earth System Dynamics before major revisions.

A few major comments are followings:

ML related

1. The setup of the supervised learning uses the combination of 18 observational datasets However, the combination of 18 observational datasets may overweight a few events and have limited difference. For instance, the events after 1980 are covered for most datasets but the events before are only covered by half of them. The authors should discuss this issue and provide additional analyses in the supplementary. One suggestion is to test with subgroup of the datasets. Another issue for the integrated observational datasets is the lack of differences for the dataset. Even though the reconstructions are all slightly different, the SSTs are still representing the same events. That is, the actual events consider in this study is only 14 CP, 20 EP, and 26 LN. This issue should be mentioned in the manuscript and needs to be tested with a small subgroup (or even extremely just one dataset) of datasets.

2. The setup of the supervised learning uses the features from 5 regions from October to March. However, limited dynamical reasons are provided and other regions and times should be mentioned (or even tested). For example, the authors show results from the smaller regions and times in the supplementary, but not larger regions and times. For instance, the north subtropical region is known to be important for the onset of CP El Niño and recent papers have found an improvement from including it (Tseng et al., 2021). And the summer is related to how specific ENSO type is onset (Yu & Fang 2018). The authors should provide dynamical reasons for the choice of the regions and times, otherwise, the study should examine more regions and times for showing the current choice is an optimal one.

Yu, J. Y., & Fang, S. W. (2018). The distinct contributions of the seasonal footprinting and charged-discharged mechanisms to ENSO complexity. *Geophysical Research Letters*, *45*(13), 6611-6618.

Tseng, Y. H., Huang, J. H., & Chen, H. C. Improving the Predictability of Two Types of ENSO by the Characteristics of Extratropical Precursors. *Geophysical Research Letters*, e2021GL097190.

Writing-related

3. The introduction is a little bit lengthy. It will be easier to read if the authors make the description more succinct. For example, the paragraph for observed CP increased (55-63) should be combined with the EP/CP introduction in the beginning. It will be great if the introduction can be better organized.

4. The ENSO complexity is recently considered with a broader perspective (Timmermann et al. 2018). Besides the EP/CP types of ENSO, the transition, propagation, and duration of ENSO are all parts of the ENSO complexity (Chen et al., 2017; Fang et al., 2020). Although these are not the focus in this paper, the ENSO complexity should be mentioned at least in the discussion section.

   Timmermann, A., An, S. I., Kug, J. S., Jin, F. F., Cai, W., Capotondi, A., ... & Zhang, X. (2018). El Niño–southern oscillation complexity. *Nature*, *559*(7715), 535-545.

   Fang, S. W., & Yu, J. Y. (2020). Contrasting transition complexity between El Niño and La Niña: observations and CMIP5/6 models. *Geophysical Research Letters*, *47*(16), e2020GL088926.

   Chen, C., Cane, M. A., Wittenberg, A. T., & Chen, D. (2017). ENSO in the CMIP5 simulations: Life cycles, diversity, and responses to climate change. *Journal of Climate*, *30*(2), 775-801.

Interpretation-related

5. The study considers the classification of CP El Niño from Pascolini-Campbell et al. (2015) for the past 120 years, which combine various CP classification methods, but no classification is applied in the multi-model large ensembles. That is, the original CP classification is not compared with the supervised learning method in the SMILEs. If the method in Pascolini-Campbell et al. (2015) is too complicated, the authors should at least choose one or two existing method to justify how the existing classification in SMILEs is different with the one from supervised learning.

6. The goal/finding is not unique for machine learning and have been discussed in studies. The authors classify ENSO events and compare the results for SMILEs. However, this can also be done by simply using existing ENSO classification method (Ng et al., 2021). The finding of this study should focus more on the uniqueness of the supervised learning. For example, since the classification method is trained from observational dataset, how each modeled ENSO in SMILEs is different with the observation? Or is machine learning do a better classification than existing methods?

   Ng, B., Cai, W., Cowan, T., & Bi, D. (2021). Impacts of low-frequency internal climate variability and greenhouse warming on El Niño–Southern Oscillation. *Journal of Climate*, *34*(6), 2205-2218.

7. The authors compare the changes of SST pattern for the EP and CP El Niño under global warming. The interpretation should be more dynamics, as this change in pattern is seldom mentioned in other studies (maybe due to the difficulty of dynamical interpretation). I will suggest the authors to eliminate this result if no dynamical explanation is provided, as this is only discussed in one paragraph (292-302). Instead, the author can focus on the change in zonal SST gradient in the mean state and compare with the frequency or amplitude.

8. The comparison of the increased CP El Niño frequency to SMILEs should be more precise. The authors use the ensemble spreads in each year to consider as the range of change for the internal variability; however, this is different with the increased CP El Niño frequency over a certain period. The authors should check how large the CP El Niño frequency can change in each ensemble and discuss the spread of the changes for all SMILEs.

Minor comments are provided below:

1. Does the training and classification use the original SST or SST anomalies? Please clearly describe in the text.
2. The calculation of frequency should also be mentioned in the method section, not only in the caption of Figure 3.
3. The Figure 6 is a bit difficult to read as there are many colors and lines.
4. Line 205, 'to far'
5. Line 48, 'niños'

---

## Author Response (AR1)

**Reviewer 1**

**general comments**

This paper integrates 18 observational datasets and machine learning algorithms (supervised classification) to classify the CP (Central Pacific), EP (Eastern Pacific), and LN (La Nina) events in the past ~120 years. The trained/tuned model was then applied to SMILEs (single model initial-condition large ensembles) to investigate both the internal variability and forced changes in each ENSO event type. The main findings from this study are 1) machine learning (ML) does a nice job in reconstructing the ENSO events in the past 2) the observed increase in the frequency of CP events after the late 1970s is within the range of internal variability in the SMILEs (thus arguing against climate change as the cause) 3) the ML algorithm doesn't project a change in CP frequency or amplitude in the following decades.

I find this paper well written, bearing important scientific merits, and nicely integrating climate model and machine learning. However, I do have several concerns and I hope the author could address them.

*We thank the reviewer for their positive and constructive comments on this manuscript.*

**specific comments**

**ML related.**

1) Metrics and scoring. The author used precision as their main metrics to check their model performance. However, as successfully detecting the CP and EP events is the most critical part, I think the author should use the recall rate. Imagine this scenario: if we have 20 total EP events, the ML successfully categorizes 5 of them to EP, and the other 15 are categorized to other types of events, and no more other events are categorized to EP, then based on the precision formula, the precision for EP will be 5/(5+0) = 1. However, the other 15 EP events are not captured by this. If using recall, then 5/(5+15) = 0.25 and it indicates the model needs to be improved. Although recall also has its own issue, I think at least a thorough explanation of why the authors chose to use precision needs to be there. And I recommend the authors compare the results of using recall compared to precision.

*Thanks for the really helpful suggestion. The recall metric has been added to all tables and in section 2.2 on lines 151-153 of the manuscript.*

2) The author used several methods to determine/evaluate the ML model (e.g., train and evaluation/test, for the train dataset, use 10-fold cross validation). I think the author also needs to explain how they tune the training model. For example (in Table 2), why they choose 1 in the KNN, why they use the specific hidden layers and max iterations in their NN. More importantly, for the random forest algorithm, the max depth seems to be too big (500). A more detailed description of how they tune (not only evaluate) the models is needed as it will change the final model structure.

*This information is provided on github in the form of jupyter notebooks (https://github.com/nicolamaher/classification). These are linked to in the code and data availability section and the caption for Table 2. We also added the following text on line 166 to better explain how the models were individually tuned. "All three algorithms hyper-parameters are tuned for optimal performance (step 6) by evaluating the performance of the algorithm using the four scores listed above at varying values of each parameter"*

3) Can the author explain why they first use HadISST as the test data set? As the author mentioned, this will cover all the events through time and is not the ideal way to evaluate the model performance. I think their second approach is more appropriate (randomly split the events across all augmentation data sets). I suggest the authors delete the HadISST part (unless I miss something…).

*This is done due to the limited number of events in the observed record. If we split via event rather than dataset we may lose events that are important to the training phase of the classification algorithm. As such we choose to keep this as is in the manuscript, but add text to better explain this choice.*

*Text on line 133 "This choice was made due to the limited number of events in the observed record, where splitting the data by even rather than dataset may result in the loss of events that are important to the training phase of the classification algorithm."*

4) The authors need to discuss whether the range of the feature values during the training will also cover the ranges for future predictions. One example is the random forest, whose prediction results will be capped by the data used for training. In a future warming world, will the features have values that are out of the scope of the current observational ones?

*This is a limitation that is now noted on line 204 "We note that if large forced changes in the SST in the tropical Pacific occur under the future scenario the algorithm will have difficulty classifying future states as it is constrained by the information provided in the observational record."*

5) In line 90, for those that don't quite know ML, I suggest the authors add a sentence or two to explain labelled dataset vs. unlabelled data.

*The following text has been added on line 88 "Supervised learning algorithms use a labelled dataset to train a classifier (e.g. observations where a class is already assigned to each individual year from previous studies). Then this classifier can be applied to unlabelled data (e.g.climate model output where no class has been assigned yet for each individual year). "*

6) line 180, "We additionally complete this split 100 times and manually choose 10 data splits that take CP and EP (the classes with the lowest numbers of events) from across the time-series, ensuring that not all events in the split come from the same part of the observational record.". I feel a bit loss here.

Does it mean the events in any split needs to cover the whole time period? Needs to be reworded or adding more details.

*Apologies for confusion this will be reworded. The algorithm sometimes chooses for example CP events that all occur during the beginning of the observational record. As data quality depends on when in the observational record the event occurs in, it is only appropriate to use data splits that include events across the time period.*

*Updated text on line 185 "We additionally complete this split 100 times and manually choose 10 data splits that take CP and EP (the classes with the lowest numbers of events) from across the time-series. This is done to ensure that not all events in the split come from the same part of the observational record. As observational data quality for SST is dependent on where in the record it occurs (e.g. lower quality early in the record) it is only appropriate to use data splits that include events across the entire record."*

7) Line 165: We have 14 CP in total (see line 110), why we only have 13 here (12/13)?

*Apologies the 2019 CP event is not included in the test dataset. This is now noted on line 174*

**Model interpretation related.**

1) A very interesting finding (and important!) from this study is that due to the interval variability of the SMILEs, the assumed change in the frequency and amplitude of the ENSO events can be covered by the models themselves (Instead of required further forcing such as climate change). This is great but I am wondering if this could simply serve as the explanation of the change in the observational trend of ENSO events. For example, in figure 3, for the CP events, the HadISST shows a significant increase in CP frequency. Although the SMILEs cover this increase, it is mainly due to the wide band between minimum and maximum, the trend by SMILE is relatively flat (or slight decrease or increase). The authors need more nuanced explanation.

*We have added an assessment of trends in individual ensemble members on line 269*

*"However, when we consider trends in individual ensemble members we get a different result. Observations show a increasing trend of 7.78(number of events per 30 years)/100year for the entire observed period (1896-2019). The two models that cover the whole period do not capture this trend (max trend MPI-GE = 5.0, CanESM5=5.0). When we consider the shorter, better observed time period of 1950 onward the trend in observations increases to 8.4. However, in this case all models that cover the entire time period are able to capture this trend (max trend MPI-GE = 15.1, CESM-LE = 13.3, CanESM2 = 11.3, GFDL-ESM2M = 19.7, CanESM5 = 11.9). This suggests that the observed increase in CP could be due to internal variability alone, as the models capture the increase in CP events in the better observed period, however the inconsistencies in the earlier period also highlight potential model biases."*

*We have also added a quantification of the max/min and mean trend from individual ensemble members to Figures 3 and 4.*

2) Line 75, The author needs to explain "undersampling internal variability" here.

*This line has been removed from the manuscript*

**technical corrections**

*All technical corrections have been applied to the revised manuscript as suggested.*

Line 40: change to "is uncertain, sparse, and intermittent" *Corrected*

Line 100: in 5. change to "to use the evaluation set to assess". In 6. Add "better" before performance *Corrected*

Line 130: change "but chose" to "we chose" *Changed to "and chose"*

Line 330: "too located too far west" reads not ideal *We removed the first instance of too*

Line 345: delete "are" before needed to evaluate ENSO *Deleted*

**Reviewer 2**

This paper applied the integrated observational dataset to train the classification of the EP El Niño, CP type El Niño, and La Niña with supervised learning and to investigate the ENSO diversity/complexity changes in multi-model large ensembles. Specifically, they found the supervised machine learning can reasonably classify ENSO events/types and the observed increase of CP El Niño events is within the range of internal variability, so does the ENSO amplitude and frequency changes. The research topic is interesting and necessary; however, there are issues in the machine learning setup and the goal/finding is not unique for machine learning. Therefore, this paper should not be accepted in Earth System Dynamics before major revisions.

A few major comments are followings:

*We thank the reviewer for their constructive comments.*

ML related

> 1.The setup of the supervised learning uses the combination of 18 observational datasets
> However, the combination of 18 observational datasets may overweight a few events and have
> limited difference. For instance, the events after 1980 are covered for most datasets but the
> events before are only covered by half of them. The authors should discuss this issue and
> provide additional analyses in the supplementary. One suggestion is to test with subgroup of the
> datasets. Another issue for the integrated observational datasets is the lack of differences for the
> dataset. Even though the reconstructions are all slightly different, the SSTs are still representing
> the same events. That is, the actual events consider in this study is only 14 CP, 20 EP, and 26
> LN. This issue should be mentioned in the manuscript and needs to be tested with a small
> subgroup (or even extremely just one dataset) of datasets.

*We have retested the algorithm with only HadISST and added the results to Table 3. However, we will continue to use this method as shown by Pascolini-Campbell et al 2015, events are classified quite differently when using different data products. As such we believe the use of many products is justified in this study.*

*We add the following text on line 189 "We additionally test the algorithms with this split for HadISST alone to test the robustness of the result to using a single dataset."*

> 2.The setup of the supervised learning uses the features from 5 regions from October to March.
> However, limited dynamical reasons are provided and other regions and times should be
> mentioned (or even tested). For example, the authors show results from the smaller regions and
> times in the supplementary, but not larger regions and times. For instance, the north subtropical
> region is known to be important for the onset of CP El Niño and recent papers have found an
> improvement from including it (Tseng et al., 2021). And the summer is related to how specific
> ENSO type is onset (Yu & Fang 2018). The authors should provide dynamical reasons for the

choice of the regions and times, otherwise, the study should examine more regions and times for showing the current choice is an optimal one.

*We have tested the addition of the North Subtropical region and including more months in the analysis. We find that including 10 months (June-March) improves the classification, while including the North Subtropical region as defined in Tseng et al, 2021 does not improve the classification, in fact it appears to degrade it, but only slightly. As such we updated the paper to use a new classification including more months, but did not include the North Subtropical Region.*

*Text has been added into the supplementary discussion in section 2.1 and table S3 to explain this choice.*

*The following text has been added to the main manuscript on line 123.*

*"Previous studies have identified that the Pacific SST in the predefined niño boxes, the north Subtropical Pacific, and the evolution of SST over the event, are the most important features for separating the two types of El Niño (e.g. Rasmusson and Carpenter, 1982; Yu and Fang, 2018; Tseng et al., 2022). In this study we use the regions niño3E (5S-5N, 120-90W), niño3W (5S-5N, 120-150W), niño4E (5S-5N, 150-170W), niño4W (5S-5N, 170-200W), and niño1.2 (0-10N, 80-90W) in each month from June to March as individual features to train the classifier (i.e. 5 regions x 10 months = 50 features). We note that we tested multiple feature sets including varying regions in the tropical and north subtropical Pacific and different sets of months for performance (see Supplementary Information 2: Supplementary Methods), and chose this set due to its high performance in the evaluation phase."*

Yu, J. Y., & Fang, S. W. (2018). The distinct contributions of the seasonal footprinting and charged‐discharged mechanisms to ENSO complexity. *Geophysical Research Letters*, *45*(13), 6611-6618.

Tseng, Y. H., Huang, J. H., & Chen, H. C. Improving the Predictability of Two Types of ENSO by the Characteristics of Extratropical Precursors. *Geophysical Research Letters*, e2021GL097190.

Writing-related

3. The introduction is a little bit lengthy. It will be easier to read if the authors make the description more succinct. For example, the paragraph for observed CP increased (55-63) should be combined with the EP/CP introduction in the beginning. It will be great if the introduction can be better organized.

*We have combined these paragraphs and revised the introduction to make it more succinct.*

4. The ENSO complexity is recently considered with a broader perspective (Timmermann et al. 2018). Besides the EP/CP types of ENSO, the transition, propagation, and duration of ENSO are all parts of the ENSO complexity (Chen et al., 2017; Fang et al., 2020). Although these are not the focus in this paper, the ENSO complexity should be mentioned at least in the discussion section.

*Discussion of this point is included on line 254 "ENSO complexity itself includes not only diversity in the form of EP and CP El Ninos, but other metrics such as the transition, propagation and duration of events (e.g. Timmermann et al., 2018, Fang & Yu, 2020, Chen et al, 2017). While the propagation of events is to some extent accounted for in the classifier with inclusion of the evolution over time, other metrics of ENSO complexity are also important in understanding ENSO events and could be considered using SMILEs in future work."*

Timmermann, A., An, S. I., Kug, J. S., Jin, F. F., Cai, W., Capotondi, A., ... & Zhang, X. (2018). El Niño–southern oscillation complexity. *Nature, 559*(7715), 535-545.

Fang, S. W., & Yu, J. Y. (2020). Contrasting transition complexity between El Niño and La Niña: observations and CMIP5/6 models. *Geophysical Research Letters*, *47*(16), e2020GL088926.

Chen, C., Cane, M. A., Wittenberg, A. T., & Chen, D. (2017). ENSO in the CMIP5 simulations: Life cycles, diversity, and responses to climate change. *Journal of Climate*, *30*(2), 775-801.

Interpretation-related

5.The study considers the classification of CP El Niño from Pascolini-Campbell et al. (2015) for the past 120 years, which combine various CP classification methods, but no classification is applied in the multi-model large ensembles. That is, the original CP classification is not compared with the supervised learning method in the SMILEs. If the method in Pascolini-Campbell et al. (2015) is too complicated, the authors should at least choose one or two existing method to justify how the existing classification in SMILEs is different with the one from supervised learning.

*Picking one method will not provide a good comparison as shown in Pascolini-Campbell (2015) the different established methods already do not agree on the classification. Including all previous methods is out of scope of this paper. We have however, included the following text in the conclusions on line 368 "Future work running all classification schemes on SMILEs and comparing this new supervised*

*learning algorithm with other methods would be informative to compare the use of different classification schemes."*

6. The goal/finding is not unique for machine learning and have been discussed in studies. The authors classify ENSO events and compare the results for SMILEs. However, this can also be done by simply using existing ENSO classification method (Ng et al., 2021). The finding of this study should focus more on the uniqueness of the supervised learning. For example, since the classification method is trained from observational dataset, how each modeled ENSO in SMILEs is different with the observation? Or is machine learning do a better classification than existing methods?

*Existing methods already do not agree in their classification of events. Unfortunately to use supervised learning we need to rely on previous classifications to label the observations. We use the results of Pascolini Campbell as this brings together a consensus from previous methods. This means that it is difficult to compare this with other methodologies. See response to point 5 above for text added to this point.*

Ng, B., Cai, W., Cowan, T., & Bi, D. (2021). Impacts of low-frequency internal climate variability and greenhouse warming on El Niño–Southern Oscillation. *Journal of Climate*, *34*(6), 2205-2218.

7. The authors compare the changes of SST pattern for the EP and CP El Niño under global warming. The interpretation should be more dynamics, as this change in pattern is seldom mentioned in other studies (maybe due to the difficulty of dynamical interpretation). I will suggest the authors to eliminate this result if no dynamical explanation is provided, as this is only discussed in one paragraph (292-302). Instead, the author can focus on the change in zonal SST gradient in the mean state and compare with the frequency or amplitude.

*We believe that keeping this is important because it provides context by comparing to other studies and it shows there is no clear relationship between amplitude change and the zonal gradient change when evaluating many SMILEs. While we don't explain the dynamics, our approach is a purely statistical exercise that makes use of more data than other studies and is therefore more robust and a valuable contribution.*

8. The comparison of the increased CP El Niño frequency to SMILEs should be more precise. The authors use the ensemble spreads in each year to consider as the range of change for the internal variability; however, this is different with the increased CP El Niño frequency over a

certain period. The authors should check how large the CP El Niño frequency can change in each ensemble and discuss the spread of the changes for all SMILEs.

*We have added an assessment of trends in individual ensemble members on line 269*

*"However, when we consider trends in individual ensemble members we get a different result. Observations show a increasing trend of 7.78(number of events per 30 years)/100year for the entire observed period (1896-2019). The two models that cover the whole period do not capture this trend (max trend MPI-GE = 5.0, CanESM5=5.0). When we consider the shorter, better observed time period of 1950 onward the trend in observations increases to 8.4. However, in this case all models that cover the entire time period are able to capture this trend (max trend MPI-GE = 15.1, CESM-LE = 13.3, CanESM2 = 11.3, GFDL-ESM2M = 19.7, CanESM5 = 11.9). This suggests that the observed increase in CP could be due to internal variability alone, as the models capture the increase in CP events in the better observed period, however the inconsistencies in the earlier period also highlight potential model biases."*

*We have also added a quantification of the max/min and mean trend from individual ensemble members to Figures 3 and 4.*

Minor comments are provided below:

*All minor comments have been addressed in the revision.*

1.Does the training and classification use the original SST or SST anomalies? Please clearly describe in the text.

*This is already described on line 119 "Before training the classifier we create anomalies by detrending the data (using a third order polynomial) and removing the seasonal cycle (mean of each month over the entire time-series after detrending)."*

2.The calculation of frequency should also be mentioned in the method section, not only in the caption of Figure 3. *This has been added on line 207-210*

3.The Figure 6 is a bit difficult to read as there are many colors and lines. *We have faded the red/blue lines and hope this makes the Figure clearer.*

4.Line 205, 'to far' *Corrected*
5.Line 48, 'niños' *No longer in the paper*

---

## Referee Report (RR1)

This paper applied the integrated observational dataset to train the classification of the EP El Niño, CP type El Niño, and La Niña with supervised learning and to investigate the ENSO diversity changes in multi-model large ensembles. Specifically, they found the supervised machine learning can reasonably classify ENSO events/types and the observed increase of CP El Niño events is within the range of internal variability after 1950 but not for the entire observational period. The research topic is interesting and necessary, and the authors have well-addressed the comments from the first review. Therefore, this paper should be accepted in Earth System Dynamics.

---

## Author Response (AR2)

**Response to review**

Reviewer 1:

I am happy to see the authors have addressed my concerns and made the text more polished. My only minor question is what does "7.78(number of events per 30 years)/100year" mean? Does it mean 7.78 events per 30 years or 7.78 events per 100 years? Please clarify.

*We apologise as this is a little confusing as both frequency and trends are defined in units per year. The frequency increases at a rate of 7.78 per 100years. Frequency is defined as events per 30years for consistency with the rest of the paper.*

*We have updated the text on line 270 to read "Observations show a increasing trend in frequency (defined as the number of events per 30years) of 7.78/100year for the entire observed period (1896-2019)"*

*We note that we have also fixed a few minor typos found on rereading the manuscript and added the following line regarding frequency to the captions of Figures 3 and 4 for clarity.*

*"We note that the trends are calculated over the entirety of the simulation length for each SMILE. This means that due to the different time periods covered, trends are not directly comparable between different SMILEs."*

---

## Author Response (AR3)

Dear Yun Li,

We note that we found two typos in the Supplement and have fixed them for the final upload.

One was a missing bracket closure ")"

The other was the line below missing from the caption of Figure S6 and S8
 "Amplitude is calculated as the November, December, January mean for the region 160E to 80W between 5N and 5S after the ensemble mean has been removed for each event in a single ensemble member taken as a running calculation along the time-series for 30-year periods."

Kind regards,
Nicola Maher